# The Potential Role of Hepatocyte Growth Factor in Degenerative Disorders of the Synovial Joint and Spine

**DOI:** 10.3390/ijms21228717

**Published:** 2020-11-18

**Authors:** Hitoshi Tonomura, Masateru Nagae, Ryota Takatori, Hidenobu Ishibashi, Tomonori Itsuji, Kenji Takahashi

**Affiliations:** Department of Orthopaedics, Graduate School of Medical Science, Kyoto Prefectural University of Medicine, Kawaramachi-Hirokoji, Kamigyo-ku, Kyoto 602-8566, Japan; sho-gun@h9.dion.ne.jp (M.N.); r-taka@koto.kpu-m.ac.jp (R.T.); h-bashi@koto.kpu-m.ac.jp (H.I.); itsuji@koto.kpu-m.ac.jp (T.I.); kenji-am@koto.kpu-m.ac.jp (K.T.)

**Keywords:** hepatocyte growth factor, osteoarthritis, intervertebral disc degeneration, articular cartilage, chondrocytes, nucleus pulposus cells, subchondral bone, vertebral endplate, peripheral nerve

## Abstract

This paper aims to provide a comprehensive review of the changing role of hepatocyte growth factor (HGF) signaling in the healthy and diseased synovial joint and spine. HGF is a multifunctional growth factor that, like its specific receptor c-Met, is widely expressed in several bone and joint tissues. HGF has profound effects on cell survival and proliferation, matrix metabolism, inflammatory response, and neurotrophic action. HGF plays an important role in normal bone and cartilage turnover. Changes in HGF/c-Met have also been linked to pathophysiological changes in degenerative joint diseases, such as osteoarthritis (OA) and intervertebral disc degeneration (IDD). A therapeutic role of HGF has been proposed in the regeneration of osteoarticular tissues. HGF also influences bone remodeling and peripheral nerve activity. Studies aimed at elucidating the changing role of HGF/c-Met signaling in OA and IDD at different pathophysiological stages, and their specific molecular mechanisms are needed. Such studies will contribute to safe and effective HGF/c-Met signaling-based treatments for OA and IDD.

## 1. Introduction

Joints are anatomical structures that are essential for flexibility, mobility, and stability. Joint diseases, therefore, lead to pain, reduced mobility, and impairments in daily activities. With the ongoing aging of society, the number of people with degenerative joint disease will continue to increase. Osteoarthritis (OA) and intervertebral disc degeneration (IDD) are the most common underlying causes of joint-related chronic disability in the elderly. OA is the most conventional type of arthritis that induces joint pain [1], and is characterized by the progressive destruction of articular cartilage, synovial inflammation, and changes in subchondral bone. IDD is one of the major causes of low back pain [2] and precedes spinal disorders, such as disc herniation, spondylosis, and lumbar spinal stenosis [3].

Although medicines, such as non-steroidal anti-inflammatory drugs, and physical therapy have been developed as treatments for degenerative joint disease, it is not yet possible to completely prevent the progression of this disease. In severe cases developed at the terminal stage of OA and IDD, surgical procedures, including total endoprosthetic replacement and spinal fusion, have to be performed. Although clinical results reveal its benefits, surgical treatment has problems in terms of a high degree of invasiveness and high economic burden [4,5]. Although the etiology of joint degeneration is complex and multifactorial, aging is clearly a risk factor for the initiation and progression of degenerative joint diseases [6,7]. With aging, there is an inevitable loss of tissue homeostasis, leading to an impaired ability of that tissue to respond to various stresses [8]. In addition, aging leads to cell death and senescence, eventual loss of multipotent stem cells, and tissue regenerative capacity [9,10]. It is imperative to identify the causes of synovial joint and spine degeneration, as this would enable more targeted and less invasive therapies aimed at keeping the elderly mobile and functional.

Hepatocyte growth factor (HGF) was first identified in 1984 as a mitogenic protein for mature hepatocytes [11,12]. HGF is a multifunctional cytokine involved in morphogenesis, cell survival, proliferation, and anti-inflammatory effects [13]. The biological effects of HGF are manifested through the specific receptor molecule, c-Met, which transduces HGF signals into the cell [14]. The HGF/c-Met system controls development and homeostasis under normal physiological conditions, and also plays an important role in protection, repair, and regeneration in various cells and tissues against injuries and disease conditions [15]. In the past three decades, various studies have revealed the potential therapeutic effects of HGF for the treatment of diverse diseases [13,15]. In this article, we provide an overview of the potential role of HGF for the treatment of degenerative joint disease and discuss the HGF/c-Met system, anticipated to be of value for identifying causes and therapies for degenerative disorders of the synovial joint and spine.

## 2. HGF/c-Met Expression in the Synovial Joint and Spine

### 2.1. Synovial Joint

#### 2.1.1. Articular Cartilage and Synovial Joint Fluid

Articular cartilage covers the diarthrodial joints and is responsible for the mechanical distribution of loads across the joints. The majority of its structure and function is controlled by chondrocytes, which regulate extracellular matrix (ECM) turnover and maintain tissue homeostasis. A previous study showed that HGF can be synthesized in rat growth plate chondrocytes [16]. Pfander et al. reported the presence and distribution of HGF and c-Met in normal and OA human articular cartilage using radioactive in situ hybridization and immunohistochemistry [17]. They showed that HGF and c-Met are expressed by chondrocytes of normal cartilage as well as OA cartilage in vivo. HGF is produced by chondrocytes of the calcified cartilage and deep zone chondrocytes next to the tidemark in normal human cartilage. c-Met is present in the calcified cartilage in the deep zone, and in a minor fraction of the superficial and mid zone chondrocytes. The synthesis of both HGF and its receptor is upregulated in OA cartilage, particularly in the deep zone. Other studies also revealed that HGF expression is found at higher levels in OA chondrocytes [18,19]. It has been suggested that upregulation of HGF may be a part of a repair mechanism in response to cell damage, but the exact role remains ambiguous. Some studies have shown that HGF and c-Met expression in chondrocytes decreased in late-stage OA (Figure 1A) [17,19]. These data indicate that the HGF/c-Met system of chondrocytes plays a role in the homeostasis and pathogenesis of human joint cartilage. It is presumed that the distribution pattern of HGF and c-Met expression in articular cartilage change according to aging and degeneration. We postulate that HGF/c-Met signaling interaction between cartilage and subchondral bone is interrupted by degeneration change in severe OA.

#### 2.1.2. Subchondral Bone

Clinical and in vitro studies have suggested that subchondral bone sclerosis due to abnormal osteoblasts is involved in the progression of OA. Human osteoblasts isolated from sclerotic subchondral OA bone tissue show an altered phenotype in vitro as well as in vivo. By investigating HGF expression in OA cartilage and subchondral bone, Guevremont et al. found that osteoblasts from the subchondral bone plate were mainly engaged in HGF synthesis [24]. Another study also identified elevated endogenous HGF levels in osteoblasts from OA subchondral bone (Figure 1A) [25]. These data suggest that HGF paracrine cross-talk between subchondral bone and cartilage may occur during OA. HGF also has significant effects on the proliferation and differentiation of osteoclast precursors, osteoclast activity, and survival [26,27]. HGF expression in osteoclasts from OA subchondral bone is yet to be studied.

### 2.2. Spine

#### Intervertebral Disc

The intervertebral disc is located between two cartilaginous end plates of the adjacent vertebra of the spine. This complex structure serves as a polyaxial joint [28]. The intervertebral disc is made of two major components: the nucleus pulposus (NP) in the center, and the annulus fibrosus (AF), which surrounds the NP [29]. Both parts play an important role in maintaining mobility and providing support to the spinal column. NP tissue consists of NP cells and ECM components, including proteoglycans and collagen type II. A previous study confirmed HGF and c-Met expression in NP cells [30]. It was shown that c-Met expression was decreased by HGF treatment and increased by inflammatory stimulation. However, the expression of HGF and c-Met in the human intervertebral disc tissue, and changes in their expression during disc degeneration were not evaluated. In addition, there were no study about the HGF/c-Met signaling in the AF and vertebral endplate (Figure 1B).

## 3. HGF/c-Met Activities in the Synovial Joint and Spine

### 3.1. Articular Cartilage Proliferation and ECM Metabolism

HGF has been found to play an important role in cartilage proliferation and ECM metabolism. A previous report showed that isolated immature rabbit and rat chondrocytes in vitro responded to HGF with cell proliferation and increased their total cell number by 1.8-fold. In addition, HGF stimulates proteoglycan synthesis in chondrocytes 1.5-fold [31]. Wakitani et al. also showed in vivo that HGF has chemotactic properties in growth plate-isolated chondrocytes, as well as those isolated from adolescent (6 months old) rabbits [32]. Moreover, to investigate articular cartilage repair using HGF in vivo, they injected HGF into rabbit knee joints, where 4-mm-diameter osteochondral defects had been made and observed the animals for 6 months. They found that HGF effectively repaired osteochondral defects. The repair process of the articular cartilage defects using HGF was shown to be much better than saline injection in all macroscopic and histologic examinations. Although the observation period was short, they concluded that HGF was one of the most promising candidates for clinical repair of articular cartilage defects. Tibesku et al. also evaluated whether intra-articular administration of HGF influences the ingrowth of osteochondral grafts in vivo using a sheep model [33]. They found that HGF positively influenced the cellularity of the transplanted osteochondral graft but could not diminish the fissures in the marginal zone of the grafts. These intra-articular injection studies have several problems, including the half-life of HGF and whether chondrocytes in the articular cartilage respond or not. It is presumed that the half-life of HGF is very short and immature cells from the synovium and periosteum also react to HGF. Another study revealed that HGF has the ability to induce both the expression and synthesis of collagenase 3 in OA chondrocytes by a kinase cascade involving SAPK/JNK [34]. In contrast, Bau et al. concluded that HGF plays a small role in cartilage ECM turnover. They also found that proteoglycan, collagenase, and aggrecanase synthesis in adult human articular cartilage was not stimulated by HGF. Although they identified c-Met m-RNA in both normal and OA cartilages, they found only low levels of HGF in these tissues (Table 1) [19]. These results suggested that HGF has the potential to promote the cell survival of chondrocytes and the articular cartilage proliferation. However, questions remain as to whether HGF stimulates the ECM metabolism. In addition, HGF/c-Met signaling is mediated by various molecular pathways, including the mitogen-activated protein kinase (MAPK), PI3K/Akt, and signal transducer and activator of transcription 3 (STAT3) pathways. The molecular mechanism by which HGF act on the articular cartilage proliferation and ECM metabolism is unknown, further studies will be needed.

### 3.2. Bony Remodeling and Osteogenesis in the Articular Joint

Grumbles et al. reported that HGF and c-Met were expressed in the growth plate of rachitic rats and in chondrocyte cultures of the proliferative zone. They found that HGF increased type II collagen synthesis and alkaline phosphatase activity, a marker of cell differentiation [16]. Interestingly, the distribution pattern of HGF is similar to the described distribution of type X collagen in OA cartilage, which has been interpreted as a molecular marker showing premature chondrocyte differentiation to hypertrophic cells [35]. Taken together, it is likely that HGF contributes to the phenotypic shift of OA chondrocytes.

Several reports have postulated that HGF influences bony remodeling and osteogenesis in OA-affected joints [16,23,25,26,27]. Dankbar et al. provided evidence of a potential role of HGF in osteophyte formation in OA by promoting monocyte chemoattractant protein-1 expression that mediates the entry of monocytes/macrophages into the OA synovium [23]. They found that levels of HGF in synovial fluid were highly correlated with the extent of osteophyte formation in OA joints. Osteophytes are considered to be a result of an attempted osteochondral repair process mediated by HGF osteogenic activity. Another potential role of HGF could be linked with its capacity to enhance bone morphogenic protein (BMP) receptors and promote fracture healing [36]. Abed et al. demonstrated that elevated endogenous HGF production in OA osteoblasts is responsible, in part, for their altered response to BMP-2 stimulation, leading to an abnormal phenotype, and is implicated in OA pathophysiology [25]. Indeed, previous reports have shown that the mechanical overload that initiates microfractures in subchondral bone provokes a biological response that potentiates the progression of articular cartilage damage in OA [37]. These data suggest that alterations in local HGF levels could promote continuous bony remodeling in OA joint tissues. Subchondral bone sclerosis is considered to be a result of the bone healing process partly mediated by HGF and BMP signaling interaction. Hence, HGF could be a potential candidate that directly alters OA osteoblast metabolism via an autocrine/paracrine pathway in OA joints.

### 3.3. Intervertebral Disc Proliferation and ECM Metabolism

NP is a non-vascular tissue and has low cell proliferative activity; therefore, self-repair of NP after degeneration is unlikely. It has been reported that in cultured rabbit NP and AF cells, HGF promotes cell proliferation under low and normal oxygen conditions [30,38]. Under hypoxic conditions, HGF treatment promoted NP cell proliferation via increased hypoxia-inducible factor-1α (HIF-1α) expression. The MAPK, PI3/Akt, and STAT3 pathways promote HIF-1α upregulation induced by HGF treatment and hypoxic stimulation [38]. Therefore, it is likely that HGF/c-Met signaling uses HIF-1α expression to regulate cell proliferation in intervertebral discs. Apoptosis has been confirmed to play a key role in the progression of IDD [39]. A previous study showed that HGF inhibited the apoptosis of NP cells induced by reactive oxygen species [30]. Inflammatory response is strongly associated with the progression of IDD [40]. Inflammatory cytokines, particularly interleukin-1β and tumor necrosis factor (TNF)-α, accelerate the progression of intervertebral disc degeneration by upregulating matrix catabolic enzymes. It was reported that increased cyclooxygenase-2, matrix metalloprotease (MMP)-3, and MMP-9 expression levels stimulated by TNF-α were suppressed by HGF treatment in NP cells [30]. An in vivo study demonstrated that HGF-loaded gel injected into the degenerative discs of rat tail models retards disc degeneration, as revealed by magnetic resonance imaging, histological, and immunohistochemical evaluation [41]. Following HGF injection into the NP, a significant trend towards an increase in T2 signal intensity, type II collagen staining in the NP, and the number of BMP-2-positive cells in the AF was observed. It is likely that HGF promotes BMP-2-induced repair of ECM degradation in degenerative discs (Table 2).

### 3.4. Neurotrophic Effect of Degenerative Spinal Disorder-Related Pain

Low back pain can arise from various sites, such as intervertebral discs, facet joints, vertebral bodies, back muscles, fascia, or sacroiliac joints [42,43,44,45]. Pain putatively originating from the intervertebral disc, often referred to as discogenic pain, is suspected to be the major source of chronic low back pain [46,47,48]. Pathological mechanisms of discogenic pain include sensory nerve ingrowth into the inner layers of the intervertebral disc, upregulation of neurotrophic factors and cytokines, and instability. Degenerative spinal disorders, such as disc herniation, spondylosis, and lumbar spinal stenosis, often trigger low back pain, as well as neurological symptoms by compressive and biochemical stresses to the peripheral nerve [49].

Weber et al. evaluated systemic biochemical factors as predictors of response to the treatment of low back pain with epidural steroid injection [50]. In this study, they found that changes in HGF levels were uniquely correlated with pain and disability in patients with spinal stenosis and degenerative disc disease. The elevated levels of HGF in degenerative spinal disorder may be due to peripheral nerve compression and disc degenerative changes [51]. These findings suggest that HGF is a candidate for the treatment of degenerative spinal disorder-related pain.

Expression of HGF and c-Met was confirmed in the peripheral nervous system [52,53]. After peripheral nerve injury, the HGF protein level was significantly increased at the injured and distal sites. c-Met expression was also markedly upregulated, almost exclusively in Schwann cells distal to the injury site [53]. HGF exhibits neurotrophic properties on different types of neurons, including motor, sensory, and parasympathetic neurons [54]. Murakami et al. demonstrated that decreases in nerve degeneration and scar tissue formation around the sciatic nerve in a constriction injury model were associated with HGF expression [55]. In another study, HGF overexpression was shown to enhance myelin thickness and axon diameter in injured nerves [53]. This result suggested that HGF/c-Met signaling plays an important role in Schwann cell-mediated nerve repair. In clinical trials, gene therapy with HGF has also been found to be safe, well tolerated, and sufficient to provide long-term symptomatic relief and improvement in quality of life for patients with painful diabetic peripheral neuropathy [56]. HGF has the potential to serve as a neurotrophic factor and promote nerve regeneration against peripheral nerve injury.

## 4. Recent Advances in Clinical Applications for Degenerative Disorder of the Synovial Joint and Spine

The articular cartilage and intervertebral discs are mostly avascular tissues, and the chondrocytes and NP cells exist in a physiologically hypoxic environment [57,58]. Once degeneration of these cells occurs, it is difficult to repair because of the limiting their own proliferative capacities. It has been shown that HGF promotes cell proliferation and survival in chondrocytes and NP cells, which is the main target of degenerative changes in OA and IDD. Therefore, HGF plays an important role in maintaining the homeostasis of articular cartilage and intervertebral discs. From our recent study [38], we suggest that HIF-1α is an intermediate of HGF/c-Met signaling for NP cell proliferation and survival. HIF-1α plays an indispensable role in the hypoxic environment of articular cartilage tissues and has been reported to be involved in the onset of OA [59]. While HGF/c-Met signaling and HIF-1α expression participate in a sort of crosstalk with each other [60,61], this phenomenon has not been fully investigated in chondrocytes and NP cells. We believe that further analysis of this mechanism will help elucidate the pathology of OA and IDD.

HGF was shown to affect ECM metabolism in chondrocytes and NP cells. However, it remains controversial whether these effects were anabolic or not. It was obvious that HGF inhibited ECM degradation of articular tissue. HGF may not have any remarkable effect on promoting ECM synthesis of cartilage cells compared to other growth factors, such as TGF, FGF, and BMP. Considering the difficulty of repairing cartilage and disc tissues at the advanced stages of degeneration, HGF/c-Met signaling-based treatments can be candidates for alternative treatment strategies, such as the prophylactic approach for cartilage degeneration.

The exact causes of OA and IDD remain elusive. However, key observations have led to the idea that abnormal tissue remodeling and disruption of tissue homeostasis are involved in structural alterations observed in cartilage and bone tissues of the OA and IDD. Clinical and in vitro studies have suggested that subchondral bone sclerosis due to abnormal local autocrine/paracrine regulation of osteoblasts is involved in the progression of OA. Because HGF has an apparent effect on bony remodeling and osteogenesis, effective HGF/c-Met signaling-based treatments for OA subchondral bone is expected to progress in the future. Recently, some studies have been conducted to elucidate the molecular mechanisms underlying the association between aberrant endplate bone remodeling and IDD [62]. Studies aimed at unraveling the changing role of HGF/c-Met signaling in endplate bone remodeling in IDD are needed.

In patients with OA and IDD, a leading cause of disability is pain, the primary symptom of the disease [1,63,64,65]. Articular cartilage and disc degeneration can be asymptomatic or symptomatic. The target of a biological therapy using HGF is a patient who has significant pain. Further study needs to be done to prove that this approach is a symptom-modifying therapy, capable of improving the pain symptoms that are associated with pathological changes. One hurdle in this approach is the lack of evidence for the origin of pain in OA and IDD. A new therapeutic approach focused on the neurotrophic effect of HGF may be potent against pain generation in OA and IDD.

## 5. Conclusions

As outlined in this review, HGF plays a key role in the development, growth, and tissue homeostasis of the synovial joint and spine. HGF/c-Met signaling has been implicated in the pathogenesis of OA and IDD, and acts in various tissues, such as cartilage, bone, and peripheral nerves. HGF is a ubiquitous growth factor that profoundly influences cell proliferation, matrix metabolism, inflammatory response, and neurotrophic action. Further work is needed to define the precise role of HGF/c-Met signaling in normal and diseased bone and joint tissues in order to determine the full potential of this versatile growth factor.

## Figures and Tables

**Figure 1 ijms-21-08717-f001:**
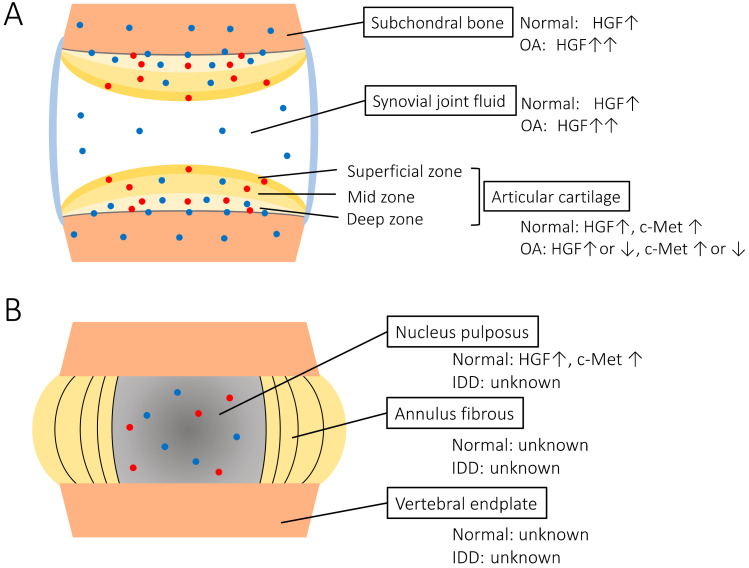
Scheme of the HGF/c-Met expression in synovial joints (**A**) and intervertebral discs (**B**) The blue dots indicate the HGF expression, and the red dots indicate the c-Met expression. OA: osteoarthritis, IDD: intervertebral disc degeneration. Several reports revealed that HGF was detected in normal and OA synovial fluid [20,21,22,23]. Moreover, the analysis of synovial fluids in OA patients revealed a positive correlation between disease severity and HGF concentrations (Figure 1A) [23]. However, the origin of HGF in the synovial fluid and whether HGF could diffuse into the articular cartilage or synovium are unknown. It is speculated that HGF acts in an autocrine or paracrine manner in the synovial joint [19,24].

**Table 1 ijms-21-08717-t001:** Effect of HGF on articular cartilage in vitro and in vivo.

Study	Cell Type and Animal Model	Effects
Takebayashi et al. (1995)	In vitro Rabbit, rat-chondrocytes	Increased cell proliferation Promotion of proteoglycan synthesis
Wakitani et al. (1997)	In vivo Rabbit-full thickness defect	Repaired with closely resembling hyaline cartilage
Reboul et al. (2001)	In vitro Human chondrocytes	Increased both the expression and synthesis of collagenase 3
Bau et al. (2004)	In vitro Human chondrocytes	No change in proteoglycan expression and synthesis No influence in the expression of collagenase and aggrecan degrading enzyme
Tibesku et al. (2011)	In vivo Sheep-osteochondral transplantation	Increased cellularity of the transplanted graft Not diminished the fissures in the marginal zone of the graft

**Table 2 ijms-21-08717-t002:** Effect of HGF on intervertebral disc in vitro and in vivo.

Study	Cell Type and Animal Model	Effects
Zou et al. (2013)	In vivo Rat-needle puncture model	Increase in NP water content Increased type II collagen staining in the NP Increased BMP-2-positive cells in the AF
Ishibashi et al. (2016)	In vitro Rabbit-NP and AF cells	Increased cell proliferation Inhibition of apoptosis induced by reactive oxygen species Suppression of MMP expression stimulated by TNF-α
Itsuji et al. (2020)	In vitro Rabbit-NP cells	Increased cell proliferation under hypoxia Upregulation of HIF-1α

NP, nucleus pulposus; BMP, bone morphogenic protein; AF, annulus fibrosus; MMP, matrix metalloproteinase; TNF, tumor necrosis factor; HIF, hypoxia-inducible factor.

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
