# Peer review of "The Potential Role of Hepatocyte Growth Factor in Degenerative Disorders of the Synovial Joint and Spine"

_ijms, 2020, doi:10.3390/ijms21228717_

Round 1
Reviewer 1 Report
This is a fairly illustrative review on a factor not yet well understood and I only have minor comments:
Lines 80-8: The paper from Bau et al (ref 22) estates that:
“Slightly higher levels of HGF were detected in chondrocytes isolated from osteoarthritic cartilage. Significant c-met expression was detected in both sample types.Conclusion. Despite the expression of its receptor c-met and its presence in articular cartilage, HGF does not appear to be a potent player in cartilage matrix turnover, at least not in terms of anabolic and catabolic gene expression in normal adult articular cartilage”
These statements somehow contradict what is said in the article under review. Please check.
Lines 106-107: The wording of the sentence should be reviewed: “Moreover, HGF in the synovial fluids of all OA patients showed increased levels of moderate and severe OA compared to the early stages of the disease”.
The tables should be mentioned in the text.
Best regards,
Author Response
Dear Reviewer,
Thank you for reviewing our manuscript and giving us very helpful suggestions. We revised the manuscript based on the reviewer’s comments; we hope our revised manuscript answers the reviewer’s questions and addresses their concerns.
Response to Comments from Reviewer
This is a fairly illustrative review on a factor not yet well understood and I only have minor comments:
Lines 80-8: The paper from Bau et al (ref 22) estates that:
“Slightly higher levels of HGF were detected in chondrocytes isolated from osteoarthritic cartilage. Significant c-met expression was detected in both sample types. Conclusion. Despite the expression of its receptor c-met and its presence in articular cartilage, HGF does not appear to be a potent player in cartilage matrix turnover, at least not in terms of anabolic and catabolic gene expression in normal adult articular cartilage”
These statements somehow contradict what is said in the article under review. Please check.
Thanks for your kind suggestion. The statements about the effect of HGF on cartilage matrix turnover were added.
INSERTED: Page 5 line 180-184 (Revised manuscript)
INSERTED: Page 5 line 185-186 (Revised manuscript)
Lines 106-107: The wording of the sentence should be reviewed: “Moreover, HGF in the synovial fluids of all OA patients showed increased levels of moderate and severe OA compared to the early stages of the disease”.
Thanks for your comment. The sentence was reviewed as below:
Page 2 line 90-91 (Revised manuscript)
“Moreover, the analysis of synovial fluids in OA patients revealed positive correlation between disease severity and HGF concentrations”.
The tables should be mentioned in the text.
As your kind suggestion, the tables were added in the text.
INSERTED: Page 5 line 184 (Revised manuscript)
INSERTED: Page 7 line 249 (Revised manuscript)
Reviewer 2 Report
I commend Tonomura et al for their attempt to synthesise and summarise the role fo HGF in musculoskeletal tissue from development to disease. However, at this time I do not believe this manuscript warrants publishing in IJMS. While I do like the breath of topics covered, more detail needs to be included. It might be suited to reduce the number of tissue covered and just focus on two or three. Additionally, a large number of the studies provided are simply cross-sectional and do not go into the mechanism of how HGF can alter MSK tissue. Lastly, the writing style is very simple and reads like a list of studies and does not provide appropriate context around each topic.Author Response
Dear Reviewer,
Thank you for reviewing our manuscript and giving us very helpful suggestions. We revised the manuscript based on the reviewer’s comments; we hope our revised manuscript answers the reviewer’s questions and addresses their concerns.
Response to Comments from Reviewer 2
I commend Tonomura et al for their attempt to synthesise and summarise the role of HGF in musculoskeletal tissue from development to disease. However, at this time I do not believe this manuscript warrants publishing in IJMS. While I do like the breath of topics covered, more detail needs to be included. It might be suited to reduce the number of tissues covered and just focus on two or three.
According to your suggestion, the authors reduced the number of tissues covered in this article.
DELETED: Page 3 line 105-119 (Revised manuscript)
2.1.3. Synovial tissue
DELETED: Page 3-4 line 133-147 (Revised manuscript)
2.2.2. Spinal cord and peripheral nerve
DELETED: Page 4 line 154-158 (Revised manuscript)
3.1. Articular joint development
DELETED: Page 6 line 194-202 (Revised manuscript)
3.3. Anti-inflammatory effect on the synovial joint
Additionally, a large number of the studies provided are simply cross-sectional and do not go into the mechanism of how HGF can alter MSK tissue. Lastly, the writing style is very simple and reads like a list of studies and does not provide appropriate context around each topic.
Thank you for your valuable suggestions. The statements about the action mechanism of HGF were added. And we modified the manuscript to provide more appropriate context.
INSERTED: Page 2 line 81-83 (Revised manuscript)
INSERTED: Page 2 line 85-88 (Revised manuscript)
INSERTED: Page 5 line 175-178 (Revised manuscript)
INSERTED: Page 5 line 184-189 (Revised manuscript)
INSERTED: Page 6 line 217-218 (Revised manuscript)
INSERTED: Page 6 line 225-227 (Revised manuscript)
INSERTED: Page 8 line 284-286 (Revised manuscript)
INSERTED: Page 8 line 289-295 (Revised manuscript)
Reviewer 3 Report
To catch the attention of the reader, please provide a drawing schematizing the site of HGF/c-Met expression in synovial joint and spine structures (sections 2.1 and 2.2).
Author Response
Dear Reviewer,
Thank you for reviewing our manuscript and giving us very helpful suggestions. We revised the manuscript based on the reviewer’s comments; we hope our revised manuscript answers the reviewer’s questions and addresses their concerns.
Response to Comments from Reviewer 3
To catch the attention of the reader, please provide a drawing schematizing the site of HGF/c-Met expression in synovial joint and spine structures (sections 2.1 and 2.2).
Thank you for your advice. A scheme of the HGF/c-Met expression in synovial joints and intervertebral discs was added as Figure 1.
INSERTED: Page 4 line 148-151 (Revised manuscript)
Figure 1. and figure legend